# An Integrative View of the Phyllosphere Mycobiome of Native Rubber Trees in the Brazilian Amazon

**DOI:** 10.3390/jof8040373

**Published:** 2022-04-06

**Authors:** Paula Luize Camargos Fonseca, Demetra Skaltsas, Felipe Ferreira da Silva, Rodrigo Bentes Kato, Giovanni Marques de Castro, Glen Jasper Yupanqui García, Gabriel Quintanilha-Peixoto, Thairine Mendes-Pereira, Anderson Oliveira do Carmo, Eric Roberto Guimarães Rocha Aguiar, Daniel Santana de Carvalho, Diogo Henrique Costa-Rezende, Elisandro Ricardo Drechsler-Santos, Fernanda Badotti, Alice Ferreira-Silva, Guilherme Oliveira, Priscila Chaverri, Aline Bruna Martins Vaz, Aristóteles Góes-Neto

**Affiliations:** 1Molecular and Computational Biology of Fungi Laboratory, Department of Microbiology, Instituto de Ciências Biológicas, Universidade Federal de Minas Gerais, Belo Horizonte 31270-901, Brazil; camargos.paulaluize@gmail.com (P.L.C.F.); thairinemp@gmail.com (T.M.-P.); 2School of Science, Technology, Engineering, & Math, Dine’ College, Tsaile, AZ 86556, USA; demetraskaltsas@gmail.com; 3Graduate Program in Bioinformatics, Universidade Federal de Minas Gerais, Belo Horizonte 31270-901, Brazil; felselva@gmail.com (F.F.d.S.); rbkato@gmail.com (R.B.K.); giomcastro@gmail.com (G.M.d.C.); glen.yupanqui@gmail.com (G.J.Y.G.); gabrielnilha@gmail.com (G.Q.-P.); ericgdp@gmail.com (E.R.G.R.A.); danielscarv@gmail.com (D.S.d.C.); 4Department of Genetics, Ecology, and Evolution, Universidade Federal de Minas Gerais, Belo Horizonte 31270-901, Brazil; anderson.o.carmo@gmail.com; 5Department of Biological Sciences, Universidade Estadual de Santa Cruz, Ilhéus 45662-900, Brazil; 6Department of Biological Sciences, Universidade Estadual de Feira de Santana, Feira de Santana 44036-900, Brazil; diogobio.dh@gmail.com; 7Department of Botany, Universidade Federal de Santa Catarina, Florianópolis 88040-900, Brazil; drechslersantos@yahoo.com.br; 8Department of Chemistry, Federal Center of Technological Education of Minas Gerais, Belo Horizonte 30421-169, Brazil; fbadotti@cefetmg.br; 9Graduate Program in Agronomy, Department of Phytotechnics and Environmental Sciences, Agricultural Sciences Center, Universidade Federal da Paraíba, Areia 58397-000, Brazil; aliceferreiradsilva@gmail.com; 10Instituto Tecnológico Vale, Nazaré, Belém 66055-090, Brazil; guilherme.oliveira@itv.org; 11Centro de Investigaciones en Productos Naturales (CIPRONA) and Escuela de Biología, Universidad de Costa Rica, San José 11501-2060, Costa Rica; pchaverr@umd.edu; 12Department of Plant Science and Landscape Architecture, University of Maryland, College Park, MD 20742, USA; 13Medical School, Universidade José do Rosário Vellano (UNIFENAS), Belo Horizonte 31270-020, MG, Brazil; alinebmv@hotmail.com

**Keywords:** metagenomics, diversity, fungi, *Hevea brasiliensis*, leaves, multiOmics

## Abstract

The rubber tree, *Hevea brasiliensis*, is a neotropical Amazonian species. Despite its high economic value and fungi associated with native individuals, in its original area in Brazil, it has been scarcely investigated and only using culture-dependent methods. Herein, we integrated in silico approaches with novel field/experimental approaches and a case study of shotgun metagenomics and small RNA metatranscriptomics of an adult individual. Scientific literature, host fungus, and DNA databases are biased to fungal taxa, and are mainly related to rubber tree diseases and in non-native ecosystems. Metabarcoding retrieved specific phyllospheric core fungal communities of all individuals, adults, plantlets, and leaves of the same plant, unravelling hierarchical structured core mycobiomes. Basidiomycotan yeast-like fungi that display the potential to produce antifungal compounds and a complex of non-invasive ectophytic parasites (Sooty Blotch and Flyspeck fungi) co-occurred in all samples, encompassing the strictest core mycobiome. The case study of the same adult tree (previously studied using culture-dependent approach) analyzed by amplicon, shotgun metagenomics, and small RNA transcriptomics revealed a high relative abundance of insect parasite-pathogens, anaerobic fungi and a high expression of *Trichoderma* (a fungal genus long reported as dominant in healthy wild rubber trees), respectively. Altogether, our study unravels new and intriguing information/hypotheses of the foliar mycobiome of native *H. brasiliensis*, which may also occur in other native Amazonian trees.

## 1. Introduction

All macroscopic eukaryotes can be considered as holobionts since they are commonly associated with a diverse microbiota comprising bacteria, archaea, and fungi in several levels of organization and interactions [1]. Plants harbor distinct communities of microorganisms that are usually specific to each plant organ or body region at the aerial parts, such as the phyllosphere [2], and the subterraneous parts, the rhizosphere [3].

While the relatively well-studied rhizosphere represents a soil–plant interface, the phyllosphere embraces the atmospheric air–plant interface [4]. This latter microhabitat is of considerable interest due to its extensive and exposed superficial area and its connection to air microbiota, which encompasses cells and other dispersal propagules of potential phytopathogens [4]. Nonetheless, most microbial communities of the phyllosphere apparently did not encompass random associations of micro-organisms deposited aerially but, instead, specific communities that jointly co-evolved with their plant hosts [1].

Phyllosphere fungi include both epiphytes (living on the leaf surface) and endophytes (living inside plant tissues); however, the distinction between epiphytic (epiphyllous or foliicolous) and endophytic fungal taxa is ambiguous [5]. Moreover, the traditional culture-dependent technique to try to exclude epiphytes by “sterilization” of the foliar surface with alcohol and sodium hypochlorite baths does not destroy fungal DNA, which may be amplified and sequenced in amplicon metagenomic sequencing [6], preventing a full separation of mycobiome in such studies. Furthermore, as endophytism may occur throughout the fungal kingdom [7], it might be understood as a life-history strategy [8], as proposed by the Viaphytism Hypothesis, and not as a rigid category [9,10]. In fact, there is also a claim that most of the fungi may potentially live, at least partially, as endophytes [8]; however, this is far from consensual in the specialized literature.

Fungal endophytes live inter- and intracellularly in distinct plant tissues [8]. Moreover, they may display an entire spectrum of ecological interactions with their plant hosts, such as positive ones (mutualism), neutral ones (commensalism), and negative ones (parasitism), besides being involved with the natural decomposition of the tissues of their plant hosts after the plant death (saprophytism) [11]. Hence, endophytic fungi are an omnipresent and phylogenetically diverse group, which may establish distinct long-term interactions with their plant hosts [12].

The rubber tree, *Hevea brasiliensis* (Willd. ex A. Juss.) Müll. Arg., is one of the 10 validly accepted species of the native neotropical and South American genus *Hevea* [13]. *Hevea brasiliensis* is one of the 227 hyperdominant trees of the Amazonia, mainly occurring in the Central and South regions of this biome but with the native individuals displaying an aggregated geographical distribution [14]. Despite being a socially important and highly economically valuable tree [15], the fungi associated with native *H. brasiliensis* individuals in its original area of occurrence in Brazil were only investigated using culture-dependent methods in two studies and only within the last three years [16,17].

Considering the methodological limitation of culture-dependent approaches, the diversity, taxonomic composition, and functions of *H. brasiliensis* mycobiome are yet to be understood in more depth. Therefore, in this study, we combined an in-silico Knowledge Discovery in Databases (KDD) approach with field and experimental approaches. The first approach consisted of an automated and thorough review of the scientific literature, host-fungal, and DNA sequence databases of fungi associated with *H. brasiliensis*. The second approach comprised regional, local, and intra-individual field sampling using metabarcoding to analyze diversity, taxonomic composition, co-occurrence, and functional ecological roles of phyllosphere mycobiomes of native rubber tree individuals in the Amazonian biome. Furthermore, in order to provide information about methodological complementarity, as a proof of concept, we also carried out a case study in one selected sampling unit to compare the fungal diversity by culture-dependent method, metabarcoding, shotgun metagenomics, and small RNA metatranscriptomics (expression profile) of the same phyllospheric fungal community.

## 2. Materials and Methods

### 2.1. In Silico Knowledge Discovery in Database (KDD) Approach

The comprehensive review of the scientific literature was performed in Scopus, Web of Science (WoS), and PubMed, and the indexing databases for the bibliographic search were addressed by an automated script written in Python 3 [18] and deposited on GitHub (https://github.com/LBMCF/search_fungi). The search in titles, abstracts, and keywords sections of the documents in the aforementioned three databases was carried out using the following search string: (“*Hevea brasiliensis*” AND fung*). Database searches included documents published from 1960 for Scopus, from 1945 for WoS, and from 1989 for PubMed to 2021 (February 14th). There was no language restriction, and the files were exported in comma-separated value (CSV) format. Duplicate records were then filtered with the Digital Object Identifier (DOI) using a format_input.py script (https://github.com/glenjasper/format-input, accessed on 23 January 2022) and removed with the remove_duplicates.py script (https://github.com/lbmcf/remove-duplicates, accessed on 23 January 2022). Subsequently, the documents were downloaded using a script that engaged the UFMG (Federal University of Minas Gerais, Belo Horizonte, MG, Brazil) network. PDF files were transformed into TXT files using the script pdf2txt.py (http://github.com/lbmcf/pdf2txt, accessed on 23 January 2022). This script internally uses the XpdfReader program (http://www.xpdfreader.com, accessed on 23 January 2022). Furthermore, all access of fungi associated with *H. brasiliensis* in USDA fungal-host (https://nt.ars-grin.gov/fungaldatabases/fungushost/fungushost.cfm, accessed on 23 January 2022) and NCBI Nucleotide (https://www.ncbi.nlm.nih.gov/nucleotide/, accessed on 23 January 2022) databases was retrieved. To search for currently accepted fungal names and remove duplicated names, we followed the data available in Index Fungorum (http://www.indexfungorum.org, accessed on 23 January 2022). Patterns of shareability and uniqueness were graphically represented by Venn diagrams [19] (in the Section 3).

### 2.2. Study Areas

The fieldwork was performed in two Brazilian Conservation Units in Eastern Amazonia: (i) Caxiuanã National Forest (CNF) and (ii) Tapajós National Forest (TNF). The permission for fieldwork was obtained from the Brazilian Ministry of Environment under the access code SISBIO 42316-2.

CNF is situated in the state of Pará, Brazil, in the Tocantins River Basin, comprising part of the municipalities of Melgaço and Portel (01°37′ S–02°15′ S; 51°19′ W–51°58′ W), with an altitude ranging from 0–80 m and a total area of 330,000 ha (Figure 1a). The vegetation is composed of 85% of non-flooded forests (*terra firme*) and 15% of seasoned or permanently flooded forests (*várzeas*, *igapós*, swamps), and the regional climate is classified as Am (Köppen system), with a mean annual temperature of 25.9 °C, a mean annual rainfall of 2011 mm, and mean annual air relative humidity of 83% [20].

TNF is situated in the state of Pará, Brazil, in the Tapajós River Basin, comprising part of the municipalities of Aveiro, Belterra, Placas, and Rurópolis (02°45′ S–04°15′ S; 54°45′ W–55°30′ W), with an altitude ranging from 19–200 m and a total area of 544,927 ha. The vegetation is composed of 93% of non-flooded forests (*terra firme*) and 7% of seasoned or permanently flooded forests (*várzeas*, *igapós*, swamps), and the regional climate is classified as Am (Köppen system), with a mean annual temperature of 25.5 °C, a mean annual rainfall of 1820 mm, and mean annual air relative humidity of 82% [21] (Figure 1b).

### 2.3. Field Sampling

The sampling strategy comprised three distinct spatial scales: regional, local, and intra-individual. The regional scale embraced the two distinct areas (TNF and CNF), which are approximately 500 km (in a straight line) away from each other, whereas the local scale encompassed each Conservation Unit per se. The intra-individual scale comprised distinct leaves from the same tree. Besides the different spatial scales, in one of the areas (TNF), two mutually excluding development stages were evaluated: adult plants and seedlings (plantlets). Furthermore, one sampling unit (tree C2 from CNF) was randomly selected as a case study in order to compare the results of a previous culture-dependent (surface-sterilized leaves) study [16]. This previous characterized culture-dependent fungal community was compared with metabarcoding, shotgun metagenomics, and small RNA transcriptomics (mycobiome expression profile). Therefore, adult individuals of *H. brasiliensis* were randomly selected from each study area (CNF and TNF). Five visually healthy compound leaves (three leaflets per leaf) with homogeneous green coloration, and without any spot, wilting, or necrotic lesions were sampled from each of the five adult individual trees. All the leaves were at D developmental stage (mature leaves) [22]. Moreover, in TNF, five plantlets of *H. brasiliensis* were also randomly selected.

### 2.4. Metabarcoding

#### 2.4.1. DNA Extraction, PCR, Metagenomics Sequencing, Bioinformatic Analyses, and Community Ecology Analyses

Leaf samples were ground with liquid nitrogen, and 300 mg of leaf tissue from the lamina (excluding petiole or middle vein) were used for genomic DNA extraction with the E.Z.N.A. Plant DNA Kit Omega Bio-Tek, according to the manufacturer’s instructions (Omega Bio-Tek, Norcross, GA, USA). The quality and quantity of DNA were evaluated using conventional agarose gel electrophoresis and spectrophotometry (NanoDrop ND 1000, NanoDrop Technologies, Wilmington, NC, USA) to guarantee a minimal load of DNA for amplification reactions (1 ng/mL). After the extraction, the nuclear ribosomal internal transcribed spacer (ITS2) region was amplified using the primers fITS7 [23] and ITS4 [24]. PCR amplification was performed using Kapa Taq DNA Polymerase High Fidelity (Roche, Cape Town, South Africa) under the following conditions: 1 initial denaturation cycle at 94 °C for 2 min, followed by 35 cycles of denaturation at 94 °C for 1 min, annealing at 60 °C for 1 min, and extension at 72 °C for 3 min, with a final extension cycle at 72 °C for 5 min. At least three independent amplification reactions were performed from the same DNA extract to account for the stochasticity. PCR products were then pooled in equimolar proportions based on their DNA concentrations and purified using AMPure beads. The DNA was quantified using a fluorescence assay using Qubit 2.0 Fluorometer (Thermo, Waltham, MA, USA) and Qubit dsDNA BR Assay Kit (Thermo, Waltham, MA, USA).

Sequencing libraries were generated using TrueSeq DNA PCR-Free Sample Preparation Kit (Illumina, San Diego, CA, USA) following the manufacturer’s recommendations, and index codes were added. The library quality was assessed on the Qubit@ 2.0 Fluorometer (Thermo Scientific, Waltham, MA, USA) and Bioanalyzer 2100 system (Agilent, Santa Clara, CA, USA). The library was sequenced on a MiSeq platform (Illumina, San Diego, CA, USA), and 2 × 250 bp paired-end reads were generated. All the raw generated sequences were deposited in NCBI SRA under accession number PRJNA767402.

The output files (FASTQ format) of the metabarcoding sequencing of each of the samples comprise our raw primary data. The bioinformatics pipeline (Appendix A) was elaborated and run on an Operational System Ubuntu 16.04.5 LTS system. The following programs were used: VSEARCH v2.9.1 [25], and BLAST v2.2.31+ [26]. Scripts in Bash [27] and Python v3.0 [18] programming languages were written to make some automatic tasks, such as merging samples or generating the abundance table. The reference database used for fungal taxonomic identification was UNITE v. 7.2 [28]. The pipeline comprised the following steps: (i) quality and length filtering was performed with VSEARCH, removing sequences smaller than 300 bp and default settings for quality filtering; (ii) dereplication was performed with VSEARCH; (iii) detection and removal of chimeric sequences was performed using the UNITE database (uchime_reference_dataset_untrimmed.fasta) and de novo implementation by VSEARCH); (iv) clustering sequences with similarity above 97% was performed with VSEARCH; (v) automatic taxonomic identification with BLASTn was done with Python-based rules (Appendix A); and (vi) generation of the abundance table was built using a Python script.

The taxonomic diversity of the phyllosphere mycobiome was analyzed both qualitatively (taxonomic composition) and quantitatively (relative taxon richness and abundance, after Hellinger transformation). Subsequently, an ordination analysis (PCoA) using Jensen–Shannon divergence was performed, and the difference between developmental stages (adults and plantlets) and collection sites (CNF-TNF) were tested with permutational analysis of multivariate dispersions (PERMDISP).

#### 2.4.2. Predicted Ecological Traits (Trophic Modes and Guilds)

Moreover, fungal taxa were functionally classified into three ecological trophic modes (saprotrophic, pathotrophic, and symbiotrophic, or a combination of these modes), and the fungal guilds within each of these trophic modes, using the FunGuild database [29], which is manually curated and referenced. For the Venn diagrams, we used the description of trophic mode gathered by FunGuild whenever possible, and manually curated scientific articles describing the trophic modes of species that were not included in this web tool.

### 2.5. Complex Networks Analyses

#### 2.5.1. Tree Similarity Network

The Bray–Curtis index was calculated to find the dissimilarity value across samples using the software Past4.0 [30]. From these values, a similarity matrix was generated by subtracting 1 and multiplying the similarity value by 100. All the similarity values were then rounded up or down to the nearest integer. The similarity matrix generated was used to construct networks based on the similarity value. The nodes of the network represent each tree from both areas (TNF and CNF) and the edges were placed between nodes depending on the similarity value between two or more nodes. For finding the network with the most relevant biological information, the network distance was calculated as described in [31]. This previous study shows that the most distant network (or critical network) is the best one to extract biological information [31]. The network distance was calculated by building a network for each similarity value ranging from 0 to 100%. For each network, two or more nodes were connected by an edge if the similarity value between two or more nodes (two or more trees) is greater than or equal to the analyzed threshold. For instance, in a network generated with a threshold of 33%, two or more nodes with a similarity value of 33 or greater are connected. Otherwise, the nodes are not connected. The network was plotted using GePhi 0.9.2 [32].

#### 2.5.2. Co-Occurrence Networks

The network was generated considering each OTU as nodes and the edges were placed between nodes when the OTUs co-occurred in one or more samples. We constructed four different networks: (a) co-occurrence network of OTUs in all the CNF samples from adult plants, (b) co-occurrence network of OTUs in all Tapajós samples from adult plants, (c) co-occurrence network of the five sampled leaves in the tree “T1”, and (d) co-occurrence network of OTUs in all the TNF samples from plantlets. TNF was the only area where one of the trees (T1) had five different leaves individually sequenced, each one corresponding to a distinct sampling unit (at an intra-individual level). For this reason, to generate the second network mentioned, a sampling unit derived from tree “T1” was randomly selected to represent this “T1” individual.

### 2.6. Case Study of a Selected Sampling Unit (C2 Tree)

#### 2.6.1. Shotgun Metagenomics

DNA extraction of the C2 tree DNA was the same as conducted for metabarcoding. A total amount of 1 µg of DNA was used for the preparation of the shotgun metagenomic library according to Illumina’s standard protocols, and the sequencing was performed on an Illumina Hiseq X Ten (Novogene Sequencing Laboratory, UC Davis Medical Center). Data preprocessing was as follows: (i) FASTQC (v0.11.4) [33] for quality analysis; (ii) Adapter removal with CUTADAPT (v. 1.18) [34]; (iii) Taxonomic assignment mapping the reads against the SILVA v138.1 SSU Ref NR99 database [35] using Kraken2 [36] with the parameter “--minimum-base-quality 5”.

#### 2.6.2. Small RNA Metatranscriptomics

Leaf fragments of the C2 tree were ground with liquid nitrogen and samples with a concentration of 500 mg were used for RNA extraction using TRIzol Reagent^®^ (Thermo Fisher Scientific, Carlsbad, CA, USA) following the manufacturer’s recommendations. The quality and quantity of RNA were evaluated using spectrophotometry (NanoDropND-1000, NanoDrop Technologies, Wilmington, DE, USA) and automated electrophoresis systems (2100 Bioanalyzer, Agilent RNA 6000 Nano Kit, Agilent Technologies, Waldbronn, DE). RNA samples were stored with 30 μL of RNA protection reagent (Omega Bio-Tek, Norcross, GA, USA). The samples were prepared for sequencing with the NEXTflex Small RNA—Seq Kit V3 and sRNAs were selected by size (15–35 nt) in denaturing SDS-PAGE electrophoresis (Bio Scientific Corp, Austin, TX, USA) and sequenced using Illumina HiSeq (Illumina, San Diego, CA, USA). Raw sequences were submitted to quality filters and adaptor removal. Sequences with low Phred quality (<20), ambiguous nucleotides, and/or a length shorter than 15 nt were eliminated. The remaining sequences were mapped and compared to all bacterial, non-fungal eukaryotes, and *H. brasiliensis* reference sequences available in the NCBI (https://www.ncbi.nlm.nih.gov/, accessed on 19 January 2022) using Bowtie, allowing one mismatch [37]. The *H. brasiliensis* genome was downloaded from the Genome Online Database—GOLD (https://gold.jgi.doe.gov, accessed on 15 January 2022). Sequences that did not present similarities with bacteria, non-fungal eukaryotes, or the plant host were used for contig assembly and subsequent analyses. Assembled contigs greater than 50 nt were characterized based on sequence similarity and pattern-based strategies. The identification of conserved domains was performed using HMMER [38].

## 3. Results

### 3.1. In Silico KDD Approach

#### 3.1.1. Literature Database

A total of 353 publicly available scientific documents with DOIs were retrieved, and the great majority of them comprised peer-reviewed articles in English published between 1968–2021, mostly concentrated in the last 15 years (Figure 2a). Most of the articles encompassed *H. brasiliensis* fungal diseases and corresponding etiological agents and their control, almost exclusively in monoculture plantations or agroforestry systems. The foliar diseases, such as South American Leaf Blight (caused by *Pseudocercospora ulei*), *Corynespora* Leaf Fall Disease (caused by *Corynespora cassiicola*), *Colletotrichum* Leaf Spot Disease (caused by *Colletotrichum gloeosporioides* species complex), and Powdery Mildew caused by *Oidium heveae* (*Erysiphe quercicola*) were reported in the retrieved articles. Root rot diseases, mainly caused by *Rigidoporus microporus*, but also by *Phellinus noxius* and *Ganoderma philippii*, were also reported. Few articles were completely devoted to the fungal diversity and ecology of arbuscular mycorrhizae and endophytes, especially in the *H. brasiliensis* natural environment and center of origin, the Amazon biome. Figure 2b displays a word cloud of the 100 most-frequent unique terms in this literature database.

#### 3.1.2. Host-Fungus Database

The USDA fungal-host database detected 285 putative species belonging to 158 fungal genera associated with *H. brasiliensis*; these are represented by widely distributed generalist genera, mainly represented by *Colletotrichum* (21 species), *Fusarium* (eight species), *Fomes* (seven species), *Hypoxylon, Marasmius*, and *Phyllosticta* (five species), followed by 10 genera with four species (*Alternaria*, *Didymella*, *Bipolaris*/*Cochliobolus*, *Diaporthe*/*Phomopsis*, *Gloeophyllum*, *Kretzschmaria*, *Nummularia*, *Periconia, Pestalotiopsis*, and *Xylaria*), 12 genera with three species, 27 genera with two species and 104 genera with a single species. The occurrence of fungal species varies spatially in different plant tissues, as well as geographically, with samples collected from 36 different countries worldwide: Asia (Brunei Darussalam, Cambodia, China, India, Indonesia, Japan, Malaysia, Myanmar, Philippines, Singapore, Sri Lanka, and Thailand), Africa (Cameroon, Central African Republic, Congo, Ivory Coast, Ghana, Liberia, Malawi, Mauritius, Nigeria, and Sierra Leone), America (Costa Rica, Cuba, Dominican Republic, Haiti, Mexico, Argentina, Bolivia, Brazil, Colombia, and Peru), and Oceania (Australia, Fiji, Papua New Guinea, and Samoa) (Figure 3a).

#### 3.1.3. Fungal DNA Sequences Associated with *H. brasiliensis*

A total 5110 fungal sequences associated with *H. brasiliensis* are deposited in the NCBI Nucleotide databases, of which 362 different taxa are identified at family level and 179 are genera. The most frequent genera are *Colletotrichum* (21), *Trichoderma* (11), *Fusarium* (10), *Trametes* (9), *Rigidoporus* (9), *Diaporthe*/*Phomopsis* (8), *Penicillium* (7), *Talaromyces* (5), and *Curvularia* (5), followed by eight genera with four and three species, 30 genera with two species, and 126 genera with only one species.

The most frequent species access associated with *H. brasiliensis* were *Corynespora cassiicola* (562), *Colletotrichum siamense* (377), *Colletotrichum fructicola* (227), *Oidium heveae* (227), *Rigidoporus microporus* (176), *Fusarium decemcellulare* (136), and *Fusarium oxysporum* (95). Also, some highly frequent hits with identification at the genus level were retrieved: *uncultured Glomus* (426), *Colletotrichum* sp. (174), *Diaporthe* sp. (113) and *Tolypocladium* sp. (102). Fifty of the retrieved genera were common to USDA fungal-host and NCBI Nucleotide databases; 108 were exclusive to the USDA database, and 130 were exclusive to NCBI database (Figure 3b). Both databases shared 52 taxa identified at the species level.

### 3.2. Field and Experimental Approach

#### 3.2.1. Metabarcoding

The network exhibited four modules (Figure 4a), suggesting that, among the samples, the nodes in the same modules would represent the most similar samples within them. Plants of the same age were not placed in the same modules (Figure 4b), and plants from the same location did not necessarily fall in the same module (Figure 4c).

At the regional scale, a total of 897 putative fungal species (OTUs) were identified as belonging to 130 genera, and a few more than one-third of them were resolved to the genus level (Appendix A). At the phylum level, *Basidiomycota* dominated the phyllosphere mycobiome when considering both the number of OTUs (65%) and incidence frequency (64%); however, the number of reads (49%) was quite similar to that of Ascomycota (51%). The three fungal orders with the highest relative richness were Agaricales (18% OTUs; 2% reads), Chaetothyriales (13% OTUs; 7% reads), and Polyporales (8% OTUs; 1% reads), but Exobasidiales (21% reads) and Microstromatales (14% reads) were far more relatively abundant when considering the number of reads. Moreover, *Meira* sp. (otus 1, 5, 17, 19, 137, 468), Ascomycota sp. 52-1 (otu 2), and *Sympodiomycopsis* (otus 3, 12, 22, 201) comprised the fungal genera with the highest relative abundance (Figure 5).

Considering the phyllospheric fungal communities of the sampling units, there was a significant difference (PERMDISP: F-value: 10.698; *p*-value: 0.017017) in the developmental stage (adults × plantlets) (Figure 6) but not for collection site (Caxiunã × Tapajós) (*p*-value > 0.05; data not shown).

Furthermore, regardless of the distinct scales (regional, local, and intra-individual) and developmental stages (adults and plantlets), a similar pattern of trophic modes and guilds has emerged considering all the different datasets. Most of the putative phyllospheric fungal species were assigned to the saprotrophic mode (51–60%), with fewer proportions for pathotrophic (8–12%), and symbiotrophic (1–3%) ecological modes, or a combination among them (Figure 7b). As the functional guilds are subsets of each main trophic mode, those related to saprotrophic mode were the most abundant in all the distinct scales and development stages: the guild represented by undefined saprotrophs and wood saprotrophs constituted the majority of the ecological guilds in the distinct mycobiomes.

Another set of networks was generated for each location (Figure 8a,b). For TNF, networks were generated for plantlet samples (Figure 8c) as well as for the different leaves of the single adult plant (Figure 8d). Overall, at the local scale, similar relative richness and abundance patterns were retrieved in each distinct study area (TNF and CNF) where Basidiomycota displayed higher relative OTU and genus richness and lower read relative abundance than Ascomycota. Nevertheless, there was a marked difference between adults and plantlets regarding the relative abundance of reads, and while most of the reads in adults were from Ascomycota, in plantlets the great majority came from Basidiomycota. Furthermore, Mucoromycota was not retrieved in plantlets. Chytridiomycota representatives were only detected in plantlets but with very low abundance. Similar to the regional scale, the fungal orders Agaricales, Chaetothyriales, Polyporales, and Exobasidiales were also recovered as the dominant ones. Nonetheless, once again, a striking difference was detected in the plantlet dataset, exhibiting more than eight-fold the read relative number of Exobasidiales, which comprised more than half of all the fungal reads retrieved from the phyllosphere mycobiome of *H. brasiliensis* plantlets. As occurred for the regional scale, amongst all the genera, *Meira* (otus 1, 5, 13, 17, 19, 137, 185, 206, 468), Ascomycota sp. 52-1 (otu 2), and *Sympodiomycopsis* (otus 3, 12, 22, 201) were those with the highest relative abundance and, again, the phyllosphere mycobiome of the plantlets showed a remarkable distinction from its adult counterpart and displayed more than half of retrieved reads associated to the genus *Meira* (otus 1, 5, 13, 17, 19, 137, 185, 206, 468) (Figure 5).

The strictest core phyllosphere community (that co-occurs in all the sampled individuals) of both adult and plantlet *H. brasiliensis* individuals in CNF and TNF comprised three taxa: Ascomycota sp. 52-1 (otu 2) and two putative distinct species of the genus *Meira* (otus 5 and 17). Interestingly, Ascomycota sp. 52-1 (otu 2) displayed negative Spearman correlations (*p* < 0.01) (−0.54 and −0.60) with the two *Meira* (otus 5 and 17) putative species, which was suggestive of a probable antagonistic interaction. Considering only the adult *H. brasiliensis* individuals in both studied areas (CNF and TNF), a more inclusive strict core community of the phyllosphere was detected, which included, as well as the aforementioned taxa, another putative *Meira* species (otu 1) and a putative species of the genus *Sympodiomycopsis* (otu 12). For each studied area (CNF and TNF), other taxa also occurred, such as a putative species of *Meyerozyma* (otu 7), for CNF and several other taxa for TNF: Ascomycota sp. 24 (otu 8), another putative species of *Meira* (otu 19), another putative species of *Sympodiomycopsis* (otu 22), *Dothideomycetes* sp. 1.2 (otu 90), and *Physisporinus* sp. (otu 90). Interestingly, the core community of plantlet phyllosphere includes, as well as exclusive taxa such as Ascomycota sp. 26 (otu 9), a taxon exclusively shared with the adults of the same collection area (TNF), Ascomycota sp. 24 (otu 8).

Despite some minor differences from the local adult datasets (TNF and CNF), the same general taxonomic composition, richness, and abundance patterns occurred at the intra-individual scale. Basidiomycota showed higher relative OTU and genus richness than the Ascomycota (Figure 8d), as well as almost equal read relative abundance. Furthermore, the orders Agaricales, Chaetothyriales, Polyporales, and Exobasidiales were also retrieved as the dominant ones, with the difference that the proportion of read relative abundance of Exobasidiales was more than double than the TNF and CNF adult plants datasets but still considerably lower (approximately three-fold) than TNF plantlets dataset. Moreover, *Meira*, Ascomycota sp. 52-1, and *Sympodiomycopsis* stood out as those with the highest relative abundance; however, the proportions of read relative abundances of *Sympodiomycopsis* displayed higher values than the TNF and CNF adult plant datasets. Nevertheless, the number of OTUs shared across different leaves was surprisingly low. This means that different leaves from the same individual plant can indeed have different fungal communities between them. Additionally, only one OTU was present in all the leaves from the same adult TNF individual (otu 2: Ascomycota sp. 52-1) (Figure 8d).

#### 3.2.2. Shotgun Metagenomics and Small RNA Metatranscriptomics of a Selected Sampling Unit (C2)

As previously analyzed using a culture-dependent approach (traditional isolation technique), one CNF adult individual (C2) was selected to be a case study in order to compare phyllospheric fungal diversity of four distinct methods: culture, metabarcoding, shotgun metagenomics, and small RNA metatranscriptomics. A total of 10 genera were isolated from the leaves of this CNF adult individual (C2 (data from: [16]), compared to 145 OTUs using metabarcoding (but only approximately 19% of them identified at genus level) and 471 OTUs in shotgun metagenomics (Appendix A). Except for a sole OTU, all of them were identified at the genus level. A few fungal genera (12) were detected by small RNA metatranscriptomics (Appendix A), and approximately 42% of them had been already detected in both culture and shotgun metagenomics. Furthermore, the genus *Trichoderma* was identified in four of the five methods used for this same sampling unit and was the third most expressed fungi besides *Fusarium* and *Hypomyces*, which were the first and second most expressed, respectively (Appendix A). Despite *Trichoderma* not having been detected in metabarcoding in this sampling unit as more than 80% of the identified OTUs were not identified at the genus level, it is plausible that one (or more) of the OTUs identified at more inclusive taxonomic categories might be related to this genus (Figure 9).

## 4. Discussion

Herein, we have associated in silico KDD with field/experimental approaches to produce an integrated view of the rubber tree phyllospheric mycobiome. A metabarcoding approach using a multiscale sampling scheme (regional, local, and intra-individual) retrieved phyllospheric core communities of all sampled individuals, adults and plantlets, and leaves of the same plant. Furthermore, taxonomic dominance patterns were distinct between adults and plantlets. On the other hand, independently of scale and developmental stage, the majority of the phyllospheric mycobiome was represented by saprotrophic taxa, and, among them were wood or undefined saprotrophs. Moreover, the case study in one selected adult sampling unit (C2) comparing mycobiome taxa (at the genus level) by traditional culture, metabarcoding, shotgun metagenomics, and small RNA metatranscriptomics retrieved that metabarcoding shared most of its identified genera with shotgun metagenomics but not with traditional culture or small RNA metatransciptomics. Conversely, shotgun metagenomics not only shared fungal genera with the three other methods for this single sampling unit (C2) but also with more than half of the fungal genera common to both NCBI and USDA databases.

In order to explain commonness and rarity in ecological communities, the concept of dividing an ecological community into core and satellite taxa was proposed [39]. Thus, core taxa would comprise those with widespread distribution and high mean local abundance, whereas satellite taxa would encompass those with restricted distribution and low mean local abundance [39]. This hypothesis was named the Core Satellite Species (CSS) hypothesis and in the last ten years, a renewed interest in the CSS hypothesis has emerged due to the rapid increase of both metabarcoding and shotgun metagenomic studies in microbial ecology, as well as the common use of the concept of the *core microbiome* [40]. Although the 100% strict cutoff of occupancy (present in all the samples) is the most commonly used definition for the core microbiome in the field of microbial ecology, other more flexible (but not so easily explainable) alternatives have also been adopted [41].

Using the aforementioned strictest occupancy definition, the core mycobiome of native *H. brasiliensis* individuals, regardless of being adults or plantlets in both 500-km distance study areas, was composed of a non-identified Ascomycota representative (otu 2) and two distinct putative species of the genus *Meira* (otu 5 and otu 17). Currently, the genus *Meira* [42] comprises six valid species: *M. argovae*, *M. geulakonigii* [43], *M. miltonrushii* [44], *M. nashicola* [45], *M. nicotianae* [46], and *M. siamensis* [47]. *Meira* species were initially described as acaropathogenic fungi [43] and were successfully applied as biological control agents against phytophagous mites [48]; however, they also occur on the surface of leaves [44,47,49], fruits [45], as endophytes [44,50], as well as in the rhizosphere [46]. The strictest core mycobiome, otu 5 and otu 17, are most similar to the phyllospheric *M. nashicola* and the acaropathogenic *M. argovae* species, respectively.

In the large and intensive monoculture *H. brasiliensis* plantations in Brazil outside the Amazon region, the infestation of phytophagous mites on both adaxial and abaxial surfaces of leaves by the species *Tenuipalpus heveae* and *Calacarus heveae* can cause severe defoliation and losses of latex production [51]. Despite their ubiquitous occurrence in native *Hevea* species in undisturbed Brazilian Amazon areas [52], the two aforementioned phytophagous mites usually display low population densities in the phyllosphere of *H. brasiliensis* plants, suggesting that their low abundance in natural areas could be related to antagonistic ecological interactions [53]. As previously explained, the genus *Meira* includes acaropathogenic species that were indeed experimentally tested (in vitro and in vivo) causing an 80–100% mortality rate in distinct phytophagous mite species [54,55]. Furthermore, one of these acaropathogenic species, *M. argovae*, which was also part of our strictest core mycobiome (otu 17), can produce argovin, a natural product that is antagonistic to mites [56]. Therefore, considering this information, we hypothesize that this core microbiome taxon (otu 17, *M. argovae*) might at least be related to the control of phytophagous mites in native *H. brasiliensis* plants in the sampled areas (CNF and TNF). Nevertheless, this hypothesis needs to be further experimentally tested.

The other taxon of the strictest core mycobiome of *H. brasiliensis* in our metabarcoding study (including the intra-individual samples) is a non-identified Ascomycota (otu 2), which is most probably related to the genus *Peltaster*, as corroborated by the comparison with the sequence database derived from the shotgun metagenomics of the C2 *H. brasiliensis* adult plant. The genus *Peltaster,* which currently comprises 15 legitimate species (according to the Mycobank database), is one of the various taxa of the so-called Sooty Blotch and Flyspeck (SBFS) fungi, a complex of many Ascomycota and Basidiomycota of non-invasive ectophytic parasites that produce superficial, dark-colored colonies on fruits, stems, and leaves of diverse plant genera [57]. Although usually associated with SBFS fungi that damage the cuticle of fruits, especially apples and pears, the basionym of this genus (*P. hedyotidis* Syd. & Syd.) was originally isolated from leaves of an Asian tropical plant (*Hedyotidis elmeri*: Rubiaceae) [58]. Additionally, one of the *Peltaster* species (*P. bertholletiae* Bat., Maia & Peres) was isolated from leaves of a Brazil nut tree (*Bertholletia excelsa*) in the Brazilian Amazon [59]. *Peltaster* species are dimorphic fungi that only live and feed on nutrients of the cuticle layer of plant leaves and fruits without killing living host cells and are therefore unconventional biotrophic fungi [57]. Since *H. brasiliensis* leaves are rich in epicuticular waxes [60] and given that *Peltaster* species have already been isolated from leaves of other Amazonian trees [59], we hypothesized that this fungal taxon of the *H. brasiliensis* strictest foliar mycobiome core is well-adapted to this ecological niche. Nevertheless, further culture-dependent analyses are still necessary to corroborate this hypothesis.

Regardless of the collection site (CNF or TNF), there were also fungal taxa that are only part of the adult *H. brasiliensis* core mycobiome, such as otu 1 (another species of *Meira* distinct from otus 5 and 17) and otu 12 (*Sympodiomycopsis*). Nowadays, the genus *Sympodiomycopsis* encompasses four species (*S. kandeliae*, *S. lanaiensis, S. paphiopedili*, and *S. yantaiensis,* according to Mycobank), all of them associated to plant and Acari. *Sympodiomycopsis* is a dimorphic Basidiomycota mainly growing in yeast state, but can also form hyphal state in culture. The genus *Sympodiomycopsis* is able to secrete a glycolipid that is supposed to be allelopathic in its native habitat, which might favor *Sympodiomycopsis* colonization and survival [61,62]. Therefore, these fungal taxa of the strictest core microbiome of *H. brasiliensis* adult individuals could be involved in the inhibition of the phyllosphere colonization by putative pathogenic fungi.

As well as in the strictest core mycobiome of *H. brasiliensis* adult individuals, exclusive co-occurrent fungal taxa were also detected in plantlet individuals, such as otu 9, which is a probable *Translucidithyrium.* This genus, as well the aforementioned *Peltaster*, is also a Sooty Blotch and Flyspeck (SBFS) fungi that has been quite recently described, and the only two known identified species are from plant leaves in tropical rainforests in southern Asia (Thailand and China) [63]. Interestingly, in spite of probably occupying the same ecological niche (leaf cuticle), the SBFS fungal taxon detected in the strictest core of *H. brasiliensis* plantlets was a completely distinct genus, which, jointly with the statistically significant different fungal communities between adults and plantlets, favors the hypothesis of the distinctness of core mycobiomes between these two life stages in *Hevea,* as previously reported [64]. Conversely, the absence of a statistically significant distinction between phyllospheric fungal communities of the two far-distant collection sites could be explained by the almost identical climate since climatic factors (especially mean annual temperature) have been recently reported as the main driving factors of the differences of *H. brasiliensis* foliar fungal communities in southern China [65].

The *H. brasiliensis* foliar mycobiome of both adults (in the two distinct study areas) and plantlets were dominated by saprotrophic fungi, especially from the guilds represented by undefined saprotrophs and wood saprotrophs. It has long been advocated that endophytism could be a life-history strategy of wood-saprotrophs [66,67]. Nonetheless, until quite recently it has not been proposed as a general and testable hypothesis. Nowadays, the Viaphytism Hypothesis (VH) states that, during scarcity times, saprotrophic fungi use leaves both as dispersal vehicles and resources [9,10]. Hence, the saprotrophic majority pattern of *H. brasiliensis* foliar mycobiome could, at least theoretically, favor the predictions of the viaphytism hypothesis [9]. The same pattern has also been detected in the foliar mycoendophytome of *Myrtus communis*, an endemic treelet of the Mediterranean biome [11].

Although restricted to one case study (an adult *H. brasiliensis* individual), the ability to compare the results of traditional culture-dependent (using the data of [16]), metabarcoding, shotgun metagenomics, and small RNA metatranscriptomics of the same plant not only has shed light on some methodological constraints of each approach but, most importantly, revealed quite surprising findings that should definitely be further explored and even experimentally tested in both the field and laboratory. Apart from expected patterns of much higher taxa richness than other approaches, the shotgun metagenomics retrieved a surprising high relative abundance of insect parasites (Ophiocordycipitaceae) and anaerobic fungi (Neocallimastigomycota), which is easily explained by the fact that it is a non-biased method and has a much higher sequencing depth [68]. The first group is well-known as fungi that adaptatively manipulate the behavior of their insect hosts; however, in recent years, there is also compelling evidence that they live, at least part of their life cycle, as endophytes [68], which could readily explain this finding. Nonetheless, the high relative abundance of Neocallimastigomycota taxa in the mycobiome of an *H. brasiliensis* adult plant was totally unexpected. Although they are usually found in the gastrointestinal tract of herbivorous mammals and reptiles [69], they have already been detected in the phyllosphere mycobiome, but usually in lower relative abundances [70]. Neocallimastygomyta species possess all the CAZYme-coding genes for lignocellulose degradation and therefore are fully capable of using plant cell wall carbohydrates for their nutrition, but they are most frequently associated with the gut of herbivorous mammals and reptiles [71]. Hence, their high representation in foliar mycobiome deserves future investigation.

Small RNA metatranscriptomics of the *H. brasiliensis* adult individual revealed that *Trichoderma* was one of the three most expressed genera. This finding is in line with a previous culture-dependent study with a huge sampling, which reported that *Trichoderma* was the dominant taxa of foliar fungal endophytes in wild *Hevea* adult trees in the non-Brazilian Amazon region [72]. Moreover, the same aforementioned authors, in a posterior study, reported that they had found a significant negative correlation between the relative abundance of *Trichoderma* and potentially pathogenic fungal endophytes [73]. Some species of *Trichoderma* that present mycoparasitism as a strategy of nutrition, such as direct hyphal interactions and the production of both enzymes and compounds of secondary metabolism [74,75], are widely used as plant disease biocontrol agents in agriculture. Taking this together, *Trichoderma* spp. are highly abundant in wild *H. brasiliensis* adult individuals and they are also actively expressing their genes as small RNA products in vivo inside their *H. brasiliensis* hosts.

## 5. Conclusions

In the current study, we tried to provide a comprehensive view of the foliar mycobiome of *H. brasiliensis*, the rubber tree, using: (i) in silico approaches, such as Knowledge Discovery in Databases (KDD) in literature, host-fungi, and molecular (DNA) databases; (ii) field and experimental approaches, such as metabarcoding in distinct areas and using different developmental stages, as well as a case study of shotgun metagenomics and small RNA metatranscriptomics of one of the adult individuals.

Quite expectedly, the scientific literature was strongly biased to fungal diseases and their control, especially in non-native ecosystems. Surprisingly, in a universe of more than 350 articles, only a few were devoted to investigate native plant individuals in their natural occurrence areas. A similar trend occurred with the host-fungus database, except for a slightly higher percentage of records of native plant individuals. Fungal DNA sequences associated with *H. brasiliensis* also reflected an intensive interest in those taxa mainly related to rubber tree fungal diseases; however, a higher richness of fungal taxa was detected, probably mainly due to the use of small-scale molecular methods (126 genera with only one species).

The metabarcoding in distinct spatial scales and developmental stages retrieved specific phyllospheric core fungal communities of all sampled individuals, adults, plantlets, and leaves of the same plant, unraveling hierarchical structured strictest core mycobiomes, in which fungal taxa of the more inclusive set (e.g., all samples) also occurred in the less inclusive sets (e.g., adults, plantlets). Interestingly, basidiomycotan yeast-like fungi, which have the potential to produce antifungal compounds, and Sooty Blotch and Flyspeck (SBFS) fungi co-occurred in all the samples and, thus, comprised the center of core mycobiomes of all the subsets.

Despite being restricted to a specific case study (a *H. brasiliensis* adult individual that has been previously studied by culture-dependent approach), shotgun metagenomics and small RNA transcriptomics uncovered (besides already expected outcomes) quite surprising and unexpected findings. Whereas shotgun metagenomics retrieved a high relative abundance of insect parasites-pathogens and anaerobic fungi, small RNA metatranscriptomics clearly revealed the high expression of small RNAs of the genus *Trichoderma*, which has long been reported in traditional isolation and culture-dependent studies as a dominant fungal taxon of healthy wild *H. brasiliensis* trees.

Altogether, our study provides intriguing and novel information and hypotheses of the foliar mycobiome of native *H. brasiliensis,* which can be further investigated in silico, in vitro, and in vivo, not only in the plant species but also in other native trees of the Amazon biome.

## Figures and Tables

**Figure 1 jof-08-00373-f001:**
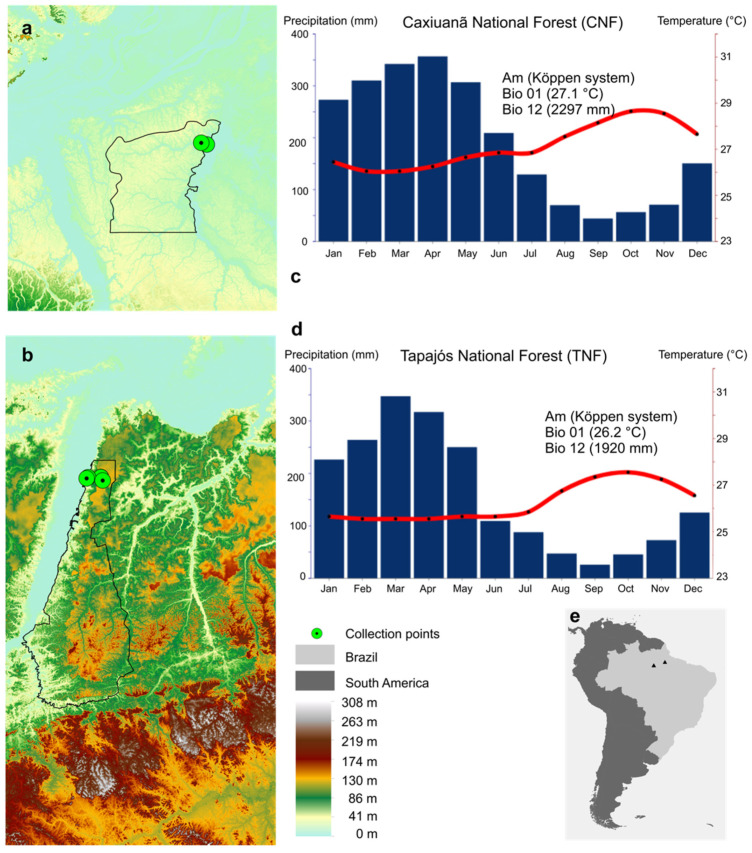
Study area. Maps show the sample collection points of two Brazilian Conservation Units in Eastern Amazonia: at the top left corner is the Caxiuanã National Forest (CNF) (**a**), and in the lower left corner is the Tapajós National Forest (TNF) (**b**). Bar charts of the precipitation (mm, blue bars) and temperature (°C, red line) to CNF (**c**) and TNF (**d**) were based on CHELSA V2.1. These cover 1981–2010 pluriannual data (~30 years) of Bio01 (annual mean temperature) and Bio12 (annual precipitation). The Digital Elevation Model (**e**) was downloaded from TOPODATA with a spatial resolution of 1 arc-sec, or roughly 30 × 30 m^2^.

**Figure 2 jof-08-00373-f002:**
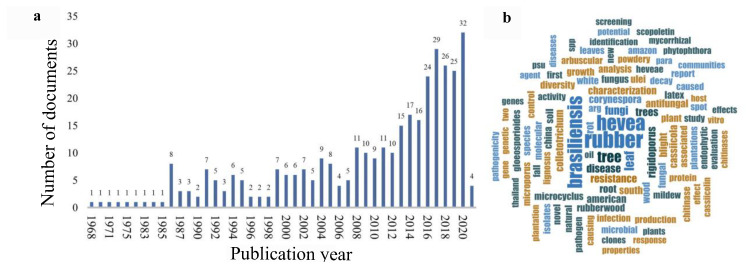
Data derived from PubMed, Scopus, and Web of Science databases: (**a**) Histogram of publications per year; (**b**) Word cloud of the 100 most-frequent unique terms by a literature search. The size of the text shows the frequency of that specific term.

**Figure 3 jof-08-00373-f003:**
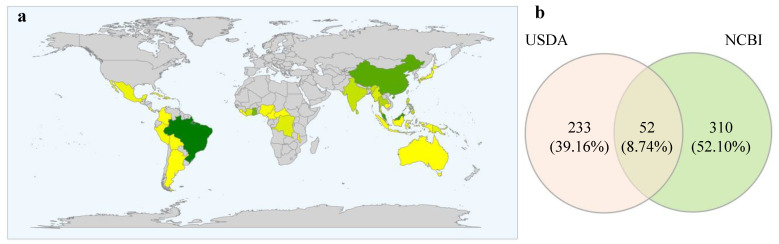
Data derived from USDA fungal-host and NCBI Nucleotide databases: (**a**) Frequency map of taxa at genus level per country, the yellow color shows low frequency and dark green color shows high frequency; (**b**) Venn diagram showing the number of shared and exclusive fungal genera between two databases.

**Figure 4 jof-08-00373-f004:**
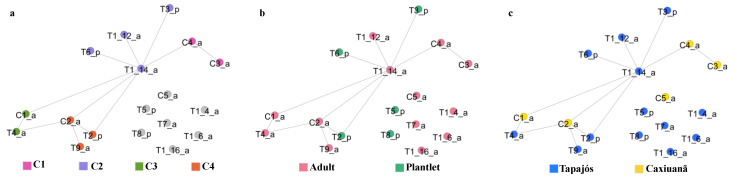
Similarity network obtained at the critical threshold of 97%, where the nodes were colored based on: (**a**) modularity calculation; (**b**) age of the plants (adult or plantlet), and (**c**) location (TNF or CNF). Nodes were connected by an edge if their similarity was greater than or equal to 97%.

**Figure 5 jof-08-00373-f005:**
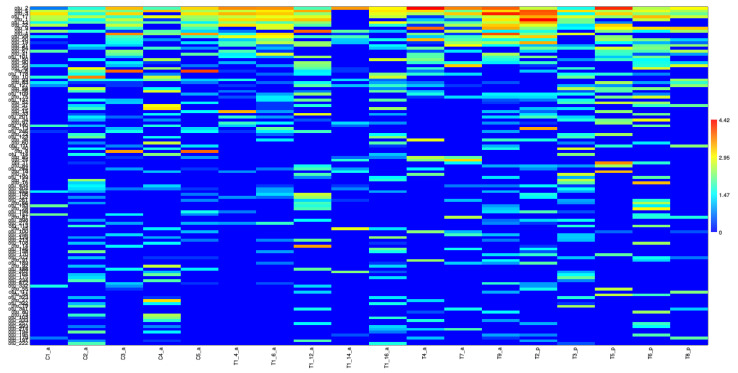
Color matrix of Hellinger-transformed abundance table of metabarcoding of the samples. For clarity, only the taxa with more than 100 reads were represented using a logarithmic (base 10) scale; correlation (Spearman) plot of Hellinger-transformed abundance table of metabarcoding of the fungal taxa. For clarity, only the first 100 OTUs were represented (which include all the core mycobiome in all combinations) (See Appendix A, for details).

**Figure 6 jof-08-00373-f006:**
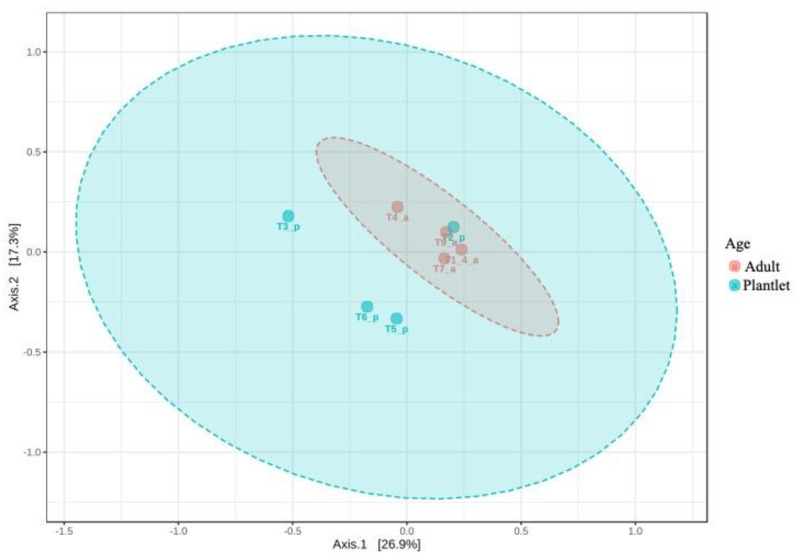
Ordination analysis (PCoA) using Jensen–Shannon divergence showing the pattern retrieved for developmental stages (adults–plantlets) of *Hevea brasiliensis* in TNF.

**Figure 7 jof-08-00373-f007:**
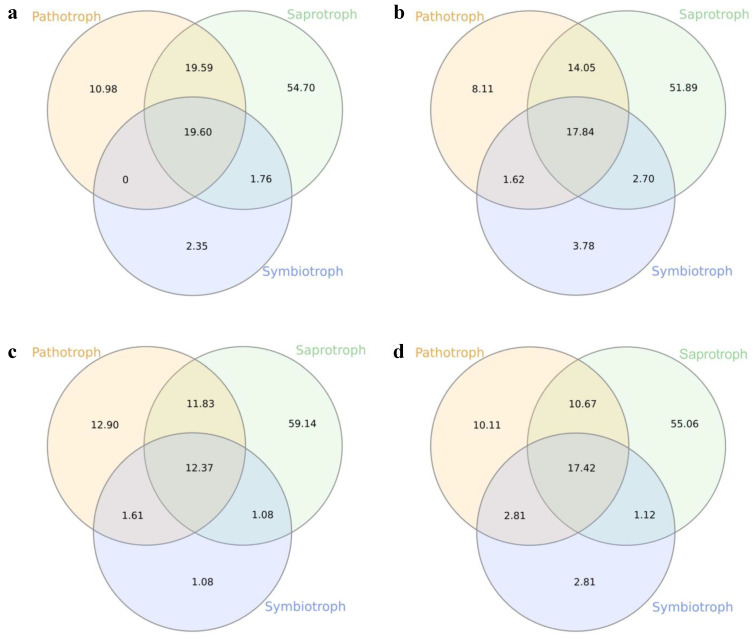
Shared and unique trophic modes among (**a**) All the samples; (**b**) CNF adults; (**c**) TNF adults; (**d**) TNF plantlets (Note: the values are in %).

**Figure 8 jof-08-00373-f008:**
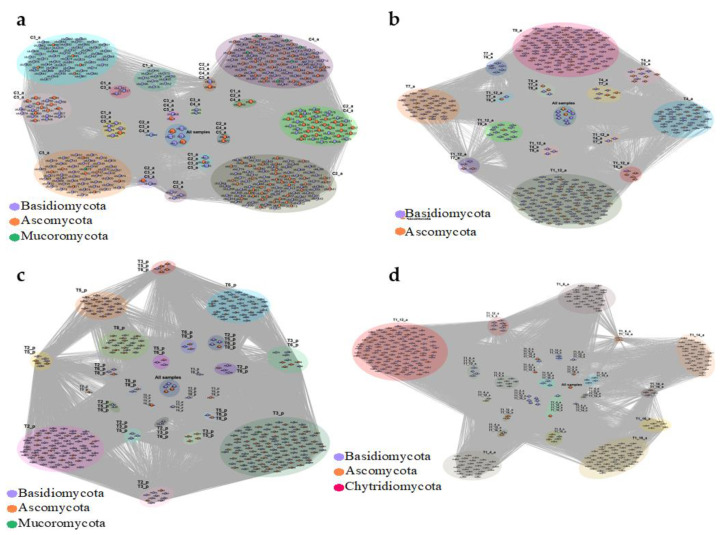
Co-occurrence network of adult plants from (**a**) CNF; (**b**) TNF (leaf T1_12_a was randomly selected to represent “T1”. See methods). Co-occurrence network of samples from TNF from (**c**) obtained from plantlets; and (**d**) obtained from five different leaves of the adult plant “T1”. Nodes represent the OTUs found, and the edges indicate that OTUs co-occurred in the same locations. The circles around the OTUs represent the sample(s) where those OTUs co-occurred.

**Figure 9 jof-08-00373-f009:**
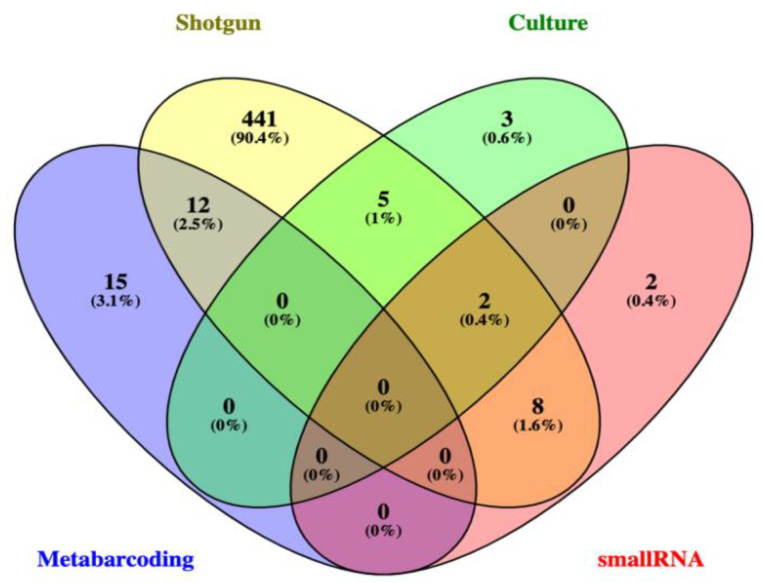
Shared and unique fungal genera in the phyllosphere of a same adult tree (CNF2) among culturing, metabarcoding, shotgun metagenomics, and small RNA transcriptomics.

## Data Availability

For In silico Knowledge Discovery in Database (KDD) approach: https://github.com/LBMCF/hevea-brasiliensis, accessed on 20 January 2022; USDA fungal-host (https://nt.ars-grin.gov/fungaldatabases/fungushost/fungushost.cfm, accessed on 20 January 2022), NCBI Nucleotide (https://www.ncbi.nlm.nih.gov/nucleotide/, accessed on 20 January 2022), and Index Fungorum databases (http://www.indexfungorum.org, accessed on 20 January 2022). For Field and experimental approach: Amplicon Metagenomics: NCBI SRA under accession number PRJNA767402; Shotgun Metagenomics: (https://github.com/LBMCF/hevea-brasiliensis/raw/main/supplementary-material/S1.zip, accessed on 31 January 2022), and Small RNA Metatranscriptomics: (https://github.com/LBMCF/hevea-brasiliensis/raw/main/supplementary-material/S2.zip, accessed on 31 January 2022).

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
