# Peer review of "An Integrative View of the Phyllosphere Mycobiome of Native Rubber Trees in the Brazilian Amazon"

_jof, 2022, doi:10.3390/jof8040373_

Round 1
Reviewer 1 Report
The contribution focuses on providing a better view of fungi that commonly associate with Havea brasiliensis in its natural environment. The authors combine a database search to query existing records, metabarcode, shotgun sequencing and metatranscriptomic tools to dissect fungal communities. Further, they compare the new data with those previously acquired using more traditional pure-culturing approaches. The new data were acquired from a total of 15 plant individuals (10 adults in two National forests and five adults and five plantlets in one of them). From each adult plant - also from plantlets (?) - five healthy leaves were sampled for this study. Further, I am also impressed by the choice of the research sites - two large National Forests in the Eastern Amazonia. One individual is selected for the shotgun and metagenome sequencing approaches as more of a proof of concept than vigorous means for hypothesis testing.
In general, the presentation in the manuscript staggers and requires further editorial attention. I did not provide any editorial suggestions beyond the introduction. I also noticed that the attention to correct grammar particularly declines in the discussion. Further, the misspelling of taxon names is rather rampant. These are difficult details to correct. I encourage the authors to be mindful about carefully checking the taxon names throughout the manuscript.
I applaud the effort put into querying the existing data for Havea-associated fungi (KDD analyses). Although perhaps not comprehensive since the earliest records in the databases were from late 1940’s, this is indeed a substantial volume of available literature that was queried. I enjoy the data presentation for the literature search. Excellent means to summarize a large volume of information. Overall, the KDD search and presentation were very clever and insightful. I tip my hat to the authors on the approach.
The means for hypothesis testing and statistical inference require some substantial attention. The authors make statements about differences that may exist among the groups of experimental units without utilizing appropriate tools for statistical inference (community analyses and comparisons among sites/plant age cohorts etc.). I encourage including commonly used tools to strengthen the inference (see below).
ADDITIONAL COMMENTS:
Lines 39-40: please soften the tone. You do not know that these are the specific taxa that produce the compounds, only that they have been assigned to taxa that are able to produce these compounds.
Line 50: consider referring to the holobiont concept, rather than the metaorganism.
Line 52: Plants harbor…
Line 53: consider “plant organ” instead of “body region”
Line 59: active cells and dispersal propagules?
Line 60: I am uncertain if this is common belief. Please rephrase. For example, you may detect AMF in the foliar samples, this would represent an aerial deposit on the surface and might not be an epiphyte or endophyte.
Line 64: Phyllosphere fungi include…
Line 70-74: meandering sentence - please rephrase. Also, I am uncomfortable with most fungi living as endophytes.
Line 75: By definition, endophytes are intracellular.
Line 87: Sparse? Maybe narrow or limited…
Line 88: fungi
Line 90: …by only two studies and only within the last three…
Line 98-99: Consider “marker sequencing” or ”metabarcoding” as the amplicon sequencing strictly speaking is not metagenomics. You have incorporated ”metabarcoding” to figures (Figure 8) later – this might be preferable.
Line 114: The abbreviation “WoS” is not defined previously in the article.
Line 128: The Venn diagrams referenced here are not included in the manuscript.
Line 176: The verbiage here seems to contrast that in line 258. Were five leaves sampled from each adult tree or only one at one site?
Line 182: and throughout: Consider “metabarcode sequencing” instead of “amplicon metagenomics”
Line 184: Clarify the amount of tissue used in your DNA extractions. Did you standardize the amount of leaf tissue in each extraction?
Line 185: Omega Bio-tek
Line 186: What was information on quality and quantity of DNA used for? Where the DNA quantities normalized for each sample?
Line 192: 35 cycles of
Line 193: When was the greater number of reactions used?
Line 195: I am not sure how the molecular weight for a size polymorphic amplicon was estimated. How were the amplicons really pooled?
Line 214: The 300bp threshold seems harsh as the fITS7-ITS4 amplicons may be shorter than this.
Line 306: Erysiphe
Lines 375-380: Please indicate the proportion of OTUs and reads per phylum - maybe also for orders.
Line 382: Microstromatales
Lines 420-426: As much as I appreciate the insight afforded by the network analyses, I strongly feel that statements about distinct communities in plantlets and adults might be best supported by more conventional statistical inference. Here and in other statements that authors make about communities being distinct among different groups, I suggest that authors robustly test their hypotheses using permutational analysis of variance analogs.
Line 429: Was the core community defined? If not, it should be.
Line 444: Which taxon might this be?
Section 3.2.2: It is not transparent earlier in the manuscript whether the authors themselves generated the culturable dataset. Please clarify whether this was derived from Vaz et al. (2018) or acquired here.
Figure 8: I applaud the use of “metabarcoding” here. Please use consistently throughout the contribution.
Lines 578-583: I am uncertain if this level of detail for the glycolipids is warranted. Note that the taxon identification in metabarcode data is uncertain as is the ability of this particular metabarcode-inferred taxon’s ability to produce these compounds.
Lines 599-604: It is true that saprobic taxa may have dominated in the leaves. However, this is far from providing evidence in support of the use of leaves as a resource during scarcity as the authors infer based on the VH.
Line 619, 621: Neocallimastigomycota
Lines 600-625: I agree the Neocallimastigomycota in the foliar samples is interesting and a puzzle. I ask the authors to entertain a thought: may this observation be in line with the large proportion of the entomopathogens in the shotgun dataset. If so, what is the inference that could be drawn from this.
Lines 637-639: The statement about plant health is a stretch. Please omit or at the very least soften the tone in this speculation.
Line 652: brasiliensis
Author Response
Reviewer 1 (Anonymous)
Comment: The contribution focuses on providing a better view of fungi that commonly associate with Hevea brasiliensis in its natural environment. The authors combine a database search to query existing records, metabarcode, shotgun sequencing and metatranscriptomic tools to dissect fungal communities. Further, they compare the new data with those previously acquired using more traditional pure-culturing approaches. The new data were acquired from a total of 15 plant individuals (10 adults in two National forests and five adults and five plantlets in one of them). From each adult plant - also from plantlets (?) - five healthy leaves were sampled for this study. Further, I am also impressed by the choice of the research sites - two large National Forests in the Eastern Amazonia. One individual is selected for the shotgun and metagenome sequencing approaches as more of a proof of concept than vigorous means for hypothesis testing.
In general, the presentation in the manuscript staggers and requires further editorial attention. I did not provide any editorial suggestions beyond the introduction. I also noticed that the attention to correct grammar particularly declines in the discussion. Further, the misspelling of taxon names is rather rampant. These are difficult details to correct. I encourage the authors to be mindful about carefully checking the taxon names throughout the manuscript.
Answer: We thank the Reviewer for his considerations to improve the text. We revised all taxa names and English language throughout the manuscript.
Comment: I applaud the effort put into querying the existing data for Hevea-associated fungi (KDD analyses). Although perhaps not comprehensive since the earliest records in the databases were from late 1940’s, this is indeed a substantial volume of available literature that was queried. I enjoy the data presentation for the literature search. Excellent means to summarize a large volume of information. Overall, the KDD search and presentation were very clever and insightful. I tip my hat to the authors on the approach.
The means for hypothesis testing and statistical inference require some substantial attention. The authors make statements about differences that may exist among the groups of experimental units without utilizing appropriate tools for statistical inference (community analyses and comparisons among sites/plant age cohorts etc.). I encourage including commonly used tools to strengthen the inference (see below).
Answer: We agreed with the Reviewer and added an ordination analysis (PCoA) using Jensen-Shannon divergence and tested the significance of the retrieved patterns with a permutational dispersion (PERMDISP) analysis to test the difference between developmental stages (adults-plantlets) (See new Figure 6).
Lines 39-40: please soften the tone. You do not know that these are the specific taxa that produce the compounds, only that they have been assigned to taxa that are able to produce these compounds.
Answer: We modified the sentence as follows: ‘Basidiomycotan yeast-like fungi that display the potential to produce antifungal compounds and a complex of non-invasive ectophytic parasites (Sooty Blotch and Flyspeck fungi) co-occurred in all samples’ (lines 39-40).
Line 50: consider referring to the holobiont concept, rather than the metaorganism.
Answer: We altered the sentence as follows: “All the macroscopic eukaryotes can be considered as holobionts since they are commonly associated to a diverse microbiota comprising bacteria, archaea, and fungi in several levels of organization and interactions’(lines 51-53).
Line 52: Plants harbor…
Answer: Modified in line 54.
Line 53: consider “plant organ” instead of “body region”
Answer: We thank Reviewer 1 for the suggestion. As some samples did not reflect specific organs of the plant (e.g.: shoots), we decided to maintain ‘body region’. We also understand that ‘body region’ is not self-explanatory, thus we added the expression ‘plant organ’ before it in line 55.
Line 59: active cells and dispersal propagules?
Answer: Modified in line 61.
Line 60: I am uncertain if this is common belief. Please rephrase. For example, you may detect AMF in the foliar samples, this would represent an aerial deposit on the surface and might not be an epiphyte or endophyte.
Answer: We agreed with Reviewer 1 and modified the sentence as follows: ‘Nonetheless, most microbial communities of the phyllosphere apparently did not encompass random associations of microorganisms aerial deposited but, instead, specific communities that jointly co-evolved with their plant hosts (lines 62-65).
Line 64: Phyllosphere fungi include…
Answer: Modified in line 66.
Line 70-74: meandering sentence - please rephrase. Also, I am uncomfortable with most fungi living as endophytes.
Answer: We completely rephrased this part, softening all the arguments and claims, as follows: ‘Furthermore, as endophytism may occur throughout the fungal kingdom, it might be understood as a life-history strategy, such as proposed by the Viaphytism Hypothesis, and not as a rigid category. In fact, there is also a claim that most of the fungi may potentially live, at least partially, as endophytes; however, this is far from consensual in specialized literature (lines 72-75).
Line 75: By definition, endophytes are intracellular.
Answer: We appreciate the consideration, and we conducted a small but extensive review on this theme to try to solve and better understand it, focused on fungal endophytes. There are many articles, mainly in the last 20 years that clearly report, using distinct microscopy techniques, that directly display hyphae of endophytic fungi within plant tissues, both intracellulaly and intercellulary (e.g.: Johnston et al., 2006; Knapp et al., 2018; Sarsaya et al., 2020, Alam et al., 2021). Therefore, we understand that it is correct to state that they occur both intra and intercellulary (lines 78-79).
Cited References:
Johnston, P. R., Sutherland, P. W., & Joshee, S. (2006). Visualising endophytic fungi within leaves by detection of (1→ 3)-ß-d-glucans in fungal cell walls. Mycologist, 20(4), 159-162. https://doi.org/10.1016/j.mycol.2006.10.003
Knapp, D.G., Németh, J.B., Barry, K. et al. Comparative genomics provides insights into the lifestyle and reveals functional heterogeneity of dark septate endophytic fungi. Sci Rep 8, 6321 (2018). https://doi.org/10.1038/s41598-018-24686-4
Sarsaiya, S.; Jain, A.; Jia, Q.; Fan, X.; Shu, F.; Chen, Z.; Zhou, Q.; Shi, J.; Chen, J. Molecular Identification of Endophytic Fungi and Their Pathogenicity Evaluation Against Dendrobium nobile and Dendrobium officinale. Int. J. Mol. Sci. 2020, 21, 316. https://doi.org/10.3390/ijms21010316
Alam B, Lǐ J, Gě Q, Khan MA, Gōng J, Mehmood S, Yuán Y, Gǒng W. Endophytic Fungi: From Symbiosis to Secondary Metabolite Communications or Vice Versa? Front Plant Sci. 2021 https://doi.org/10.3389/fpls.2021.791033
Line 87: Sparse? Maybe narrow or limited…
Answer: We would want to say that the native H. brasiliensis individuals display an aggregated distribution throughout the Amazon biome. The sparsity is related to the distance among the aggregated (or clustered) individuals. We rewrote to make this statement clearer (line 90).
Line 88: fungi
Answer: Modified in line 91.
Line 90: …by only two studies and only within the last three…
Answer: Modified in line 93-94.
Line 98-99: Consider “marker sequencing” or ”metabarcoding” as the amplicon sequencing strictly speaking is not metagenomics. You have incorporated ”metabarcoding” to figures (Figure 8) later – this might be preferable.
Answer: We opted to change to ‘metabarcoding’, as suggested (lines 102 and 107).
Line 114: The abbreviation “WoS” is not defined previously in the article.
Answer: We are sorry for the inconvenience. ‘WoS’ is the abbreviation of the Web of Science database, which is mentioned in line 113. Therefore, we added this information in line 113.
Line 128: The Venn diagrams referenced here are not included in the manuscript.
Answer: We apologize for the mistake. The reference cited here is about the tool used to build the diagrams. Venn diagram related to this section is in Figure 7 (line 415) in the Results section. We added this information in line 132.
Line 176: The verbiage here seems to contrast that in line 258. Were five leaves sampled from each adult tree or only one at one site?
Answer: Thank you for indicating it for us. In fact, there was a mistake in line 258. We corrected it in this new version o four manuscript (lines 282-286): TNF was the only area where one of the trees (T1) had five different leaves individually sequenced, each one corresponding to a distinct sampling unit (at intra-individual level). For this reason, to generate the second network mentioned, one of these sampling units derived from tree “T1” was randomly selected to represent this “T1” individual.
Line 182: and throughout: Consider “metabarcode sequencing” instead of “amplicon metagenomics”
Answer: We thank the Reviewer for the suggestion and modified the term accordingly throughout the manuscript.
Line 184: Clarify the amount of tissue used in your DNA extractions. Did you standardize the amount of leaf tissue in each extraction?
Answer: Sorry for not clarifying it in the text. We standardized it by always using 300 mg of leaf tissue from the lamina (excluding petiole or middle vein) in all the samples (lines 192-193).
Line 185: Omega Bio-tek
Answer: Modified in line 194 and 195.
Line 186: What was information on quality and quantity of DNA used for? Where the DNA quantities normalized for each sample?
Answer: We rewrote this part (Wetlab Metabarcoding Methods) and explain in detail in the text (Please see Lines 195-198)
Line 192: 35 cycles of
Answer: Modified in lines 202-203.
Line 193: When was the greater number of reactions used?
Answer: We performed three independent PCR technical replicates in all the samples (lines 204-206).
Line 195: I am not sure how the molecular weight for a size polymorphic amplicon was estimated. How were the amplicons really pooled?
Answer: It was a mistake. They were pooled based only on DNA concentration. It was corrected in the text (Line 207).
Line 214: The 300bp threshold seems harsh as the fITS7-ITS4 amplicons may be shorter than this.
Answer: We used this threshold in a conservative way since, at least in some taxa, they may reach around 300 bp.
Line 306: Erysiphe
Answer: Modified in line 334.
Lines 375-380: Please indicate the proportion of OTUs and reads per phylum - maybe also for orders.
Answer: As suggested, we indicate all the proportions of OTUs and reads per phylum and Order) (see lines 409-413).
Line 382: Microstromatales
Answer: Modified in line 413.
Lines 420-426: As much as I appreciate the insight afforded by the network analyses, I strongly feel that statements about distinct communities in plantlets and adults might be best supported by more conventional statistical inference. Here and in other statements that authors make about communities being distinct among different groups, I suggest that authors robustly test their hypotheses using permutational analysis of variance analogs.
Answer: We agreed with the Reviewer and added an ordination analysis (PCoA) using Jensen-Shannon divergence and tested the significance of the retrieved patterns with a permutational analysis of multivariate dispersions (PERMDISP) analysis to test the difference between developmental stages (adults-plantlets) and collection sites (CNF-TNF) (See figure 6 – Lines 426-429).
Line 429: Was the core community defined? If not, it should be.
Answer: The core community was defined as the community ‘that co-occurs in all the sampled individuals. We changed the place of this information to improve clarity to line 485. We also better explained the core-satellite hypothesis in discussion (lines 639-647).
Line 444: Which taxon might this be?
Answer: Sorry, we forgot to include this information: Ascomycota sp. 24 (otu 8). We inserted it in the text (line 485).
Section 3.2.2: It is not transparent earlier in the manuscript whether the authors themselves generated the culturable dataset. Please clarify whether this was derived from Vaz et al. (2018) or acquired here.
Answer: We mentioned in line 179 that the data from culture-dependent organisms are from Vaz et al., 2018. To improve clarity, we added the following sentence: ‘This previously characterized culture-dependent fungal community was compared with metabarcoding, shotgun metagenomics, and small RNA transcriptomics (mycobiome expression profile) (lines 179-180).
Cited reference:
Vaz, A.B.M.; Fonseca, P.L.C.; Badotti, F.; Skaltsas, D.; Tomé, L.M.R.; Silva, A.C.; Cunha, M.C.; Soares, M.A.; Santos, V.L.; Oliveira, G.; Chaverri, P.; Góes-Neto, A. A multiscale study of fungal endophyte communities of the foliar endosphere of native rubber trees in Eastern Amazon. Sci. Rep. 2018, 8, 1–11. https://doi.org/10.1038/s41598-018-34619-w
Figure 8: I applaud the use of “metabarcoding” here. Please use consistently throughout the contribution.
Answer: We agreed with the Reviewer and modified the text accordingly.
Lines 578-583: I am uncertain if this level of detail for the glycolipids is warranted. Note that the taxon identification in metabarcode data is uncertain as is the ability of this particular metabarcode-inferred taxon’s ability to produce these compounds.
Answer: We softened the tone of the sentence and modified it, as follows: ‘The genus Sympodiomycopsis is able to secrete a glycolipid (more specific, a cellobiose lipid) that is supposed to be allelopathic in native habitat, which might favor Sympodiomycopsis colonization and survival [61, 62] (Lines 625-629).
Lines 599-604: It is true that saprobic taxa may have dominated in the leaves. However, this is far from providing evidence in support of the use of leaves as a resource during scarcity as the authors infer based on the VH.
Answer: We softened the tone of this part of the text (lines 655-656).
Line 619, 621: Neocallimastigomycota
Answer: Modified in lines 669, 674, 678.
Lines 600-625: I agree the Neocallimastigomycota in the foliar samples is interesting and a puzzle. I ask the authors to entertain a thought: may this observation be in line with the large proportion of the entomopathogens in the shotgun dataset. If so, what is the inference that could be drawn from this.
Answer: This is an extremely interesting question and, perhaps, a future investigation to deeply explore it will be very fruitful. Until very recently, Neocallimastigomycota was considered restricted to the guts of mammals, but many microbiome studies have been revealing unexpected distinct ecological niches, such as, for instance, phyllospheres. Maybe the high relative abundance of Neocallimastigomycota is really associated with the high relative abundance. Our hypothesis is that these strictly anaerobic fungi could also occur in anaerobic sites in the inner parts of insect bodies, perhaps even in their guts. Unfortunatelly, we have not found any strong evidence in literature to corroborate this hypothesis but we think it is quite probable.
See reference for example: Liu, L., Lu, L., Li, H., Meng, Z., Dong, T., Peng, C., & Xu, X. (2021). Divergence of phyllosphere microbial communities between females and males of the dioecious Populus cathayana. Molecular Plant-Microbe Interactions, 34(4), 351-361. DOI: https://doi.org/10.1094/MPMI-07-20-0178-R
Lines 637-639: The statement about plant health is a stretch. Please omit or at the very least soften the tone in this speculation.
Answer: As suggested, we suppressed this statement (lines 696-697).
Line 652: brasiliensis
Answer: Modified in line 700.
Reviewer 2 Report
In the manuscript ‘An integrative view of phyllosphere mycobiome of native rubber trees in the Brazilian Amazon’, the authors analyzed fungal community inhabiting leaves of the rubber tree, Hevea brasiliensis, by integrated approach including literature research, metagenomics and small RNA metatranscriptomics. The authors identified core mycobiome of several species, epi- and endophytic fungi, distinctive for healthy plants.
In my opinion, the topic analyzed here, is of interest for the readers of Journal of Fungi as an important study of phyllosphere mycobiome. I have only a few remarks to improve the clarity of the manuscript, as listed below:
It is worthy to discuss an important information about chemical composition of the leaf surface to explain the reason why Basidiomycota are dominant phylum over Ascomycota and why particular species occupy the leaf surface. Is it determined by special cuticle composition, waxes or carbohydrates as a carbon source for these fungi? Or the plant secretes other compounds, attractive for Basidiomycota? How it looks in comparison to the leaf surface of other plants?
Row 532: citation format
Supplementary materials should be uploaded on the JoF submission dashboard, for convenience
Suppl.2 consists of genus names only, without metatranscriptomics data, as authors described
Author Response
General Comment: In the manuscript ‘An integrative view of phyllosphere mycobiome of native rubber trees in the Brazilian Amazon’, the authors analyzed fungal community inhabiting leaves of the rubber tree, Hevea brasiliensis, by integrated approach including literature research, metagenomics and small RNA metatranscriptomics. The authors identified core mycobiome of several species, epi- and endophytic fungi, distinctive for healthy plants.
In my opinion, the topic analyzed here, is of interest for the readers of Journal of Fungi as an important study of phyllosphere mycobiome. I have only a few remarks to improve the clarity of the manuscript, as listed below:
Comment 1: It is worthy to discuss an important information about chemical composition of the leaf surface to explain the reason why Basidiomycota are dominant phylum over Ascomycota and why particular species occupy the leaf surface. Is it determined by special cuticle composition, waxes or carbohydrates as a carbon source for these fungi? Or the plant secretes other compounds, attractive for Basidiomycota? How it looks in comparison to the leaf surface of other plants?
Answer: Unfortunately, the literature on the significant associations between the chemical composition of leaf surfaces of native Neotropical trees and their microbiome communities are practically restricted to the bacterial component of these communities. We were not able to find any studies that robustly support, in a conclusive manner, this association. Nevertheless, there is a study that suggests why Basidiomycota unicellular fungi (yeasts) could probably be dominant in arboreal phyllosphere (Voříšková,; Baldrian, 2013). In this study, the authors carried out a time-series study of fungal communities to evaluate (among other aspects) the involvement of phyllosphere fungi in the decomposition of the leaves of a typical temperate forest tree. Phyllosphere fungi are advantageous in gaining access to readily available nutrients in live leaves and later ( after senescence) to the dead leaf biomass. At least for adult individuals of native Hevea brasiliensis, the leaves were all collected at D developmental stage (mature leaves), which means a stage immediately before the senescence, and possibly richer in Basidiomycota yeasts. Furthermore, another more recent study also points out this same possibility (Koivusaari et al., 2019).
References cited: Voříšková, J., & Baldrian, P. (2013). Fungal community on decomposing leaf litter undergoes rapid successional changes. The ISME journal, 7(3), 477-486. https://doi.org/10.1038/ismej.2012.116
Koivusaari, P., Tejesvi, M. V., Tolkkinen, M., Markkola, A., Mykrä, H., & Pirttilä, A. M. (2019). Fungi originating from tree leaves contribute to fungal diversity of litter in streams. Frontiers in microbiology, 10, 651. https://doi.org/10.3389/fmicb.2019.00651
Row 532: citation format
Answer: Modified in line 578.
Supplementary materials should be uploaded on the JoF submission dashboard, for convenience
Answer: We apologize for the inconvenience. We added the supplementary materials to the dashboard in this new version.
Suppl.2 consists of genus names only, without metatranscriptomics data, as authors described
Answer: We apologize for the inconvenience. We added the complete table (with all the variables) in this new version.
Reviewer 3 Report
The authors studied the phyllosphere mycobiome of native rubber trees in the Brazilian Amazon. Overall, the manuscript is well written and suitable for publication.
Minor changes and corrections include:
What does the authors refer to when saying "adult individual" Line 35?
The authors stated in the aims that the want to "study, we combined an in silico Knowledge Discovery in Databases (KDD) approach with field and experimental approaches". But why do they want to do this?
The authors needs to do a grammar check. For example, "PCR amplification was performed using Kapa Taq DNA Polymerase High Fidelity Roche, Cape Town, South Africa)" (Lines 189-190), should read: "PCR amplification was performed using Kapa High Fidelity Taq DNA Polymerase (Roche, Cape Town, South Africa)" .
Also, "VSEARCH v2.9.1 [25](Rognes et al.," Line 209), should read VSEARCH v2.9.1 [25].
USDA fungal-host and NCBI Nucleotide databases needs to be referenced.
Can the authors more clearly explained the section about trophic mode? How did they get the numbers represented in the Venn diagrams?
Figure 7 is very difficult to read.
Author Response
The authors studied the phyllosphere mycobiome of native rubber trees in the Brazilian Amazon. Overall, the manuscript is well written and suitable for publication.
Minor changes and corrections include:
Comment: What does the authors refer to when saying "adult individual" Line 35?
Answer: The case study of shotgun metagenomics and smallRNA metatranscriptomics was carried out, as a proof of concept, in only one sampling unit: an adult tree individual (named C2) in the CNF (Caxiuana National Forest) collecting site, since this same tree was one of those previously studied using culture-dependent approach by our reseach group (Please see: Vaz, A.B.M.; Fonseca, P.L.C.; Badotti, F.; Skaltsas, D.; Tomé, L.M.R.; Silva, A.C.; Cunha, M.C.; Soares, M.A.; Santos, V.L.; Oliveira, G.; Chaverri, P.; Góes-Neto, A. A multiscale study of fungal endophyte communities of the foliar endosphere of native rubber trees in Eastern Amazon. Sci. Rep. 2018, 8, 1–11. https://doi.org/10.1038/s41598-018-34619-w).
Comment: The authors stated in the aims that the want to "study, we combined an in silico Knowledge Discovery in Databases (KDD) approach with field and experimental approaches". But why do they want to do this?
Answer: The main idea is to review all the fungi associated with Hevea brasiliensis based on published papers, host-fungus, and nucleotide databases, analyze the retrieved patterns, and used them to discuss and compared with our primary data based on metabarcoding, metagenomics and metatranscriptomics.
Comment: The authors needs to do a grammar check. For example, "PCR amplification was performed using Kapa Taq DNA Polymerase High Fidelity Roche, Cape Town, South Africa)" (Lines 189-190), should read: "PCR amplification was performed using Kapa High Fidelity Taq DNA Polymerase (Roche, Cape Town, South Africa)" .
Answer: We thank Reviewer 3 for his considerations to improve the text. We revised the text to improve the quality of grammar, English language, and style throughout the manuscript.
Comment: Also, "VSEARCH v2.9.1 [25](Rognes et al.," Line 209), should read VSEARCH v2.9.1 [25].
Answer: Modified in line 222.
Comment: USDA fungal-host and NCBI Nucleotide databases needs to be referenced.
Answer: They are now referenced in this new version (Lines: 764-766)
Comment: Can the authors more clearly explained the section about trophic mode? How did they get the numbers represented in the Venn diagrams?
Answer: FUNGuild (http://www.funguild.org/) is a web tool that describes trophic modes for many fungal species. The description of such trophic modes is manually curated and referenced by the authors (please see the reference below) and is available in https://github.com/UMNFuN/FUNGuild, for the source code of this tool, and http://www.stbates.org/funguild_db.php, for the manually curated database, which currently includes 22,726 taxons. For our Venn diagrams we used the description of trophic mode gathered by FUNGuild whenever possible, and manually curated scientific articles describing their trophic mode in the case of species not included in this web tool. Each species might present one or more associated trophic modes, for example, one species can be both a Saprotrophic and a Pathotrophic fungus. In our analysis, species presenting these trophic modes would appear in the interface of these two circles. To improve clarity, we added this information in lines 251-254.
Comment: Figure 7 is very difficult to read.
Answer: We acknowledge the reviewer's concern about the figure quality and we agree. Because the networks have a large number of nodes and labels, making the labels bigger can also make the figure very messy and hard to read. For this reason, we increased the figure quality to 600 DPI. We realized that inserting the figure in the .docx file decreased the figure definition. Therefore, please find the figure in .pdf attached to the dashboard. Additionally, this figure is now Figure 8.